# STRUCTURE-AWARE ATTENTION BASED ON VECTOR SYMBOLIC ARCHITECTURES

## ABSTRACT

The introduction of the Transformer has brought about a revolution in AI. Central to the success of the Transformer architecture is the self-attention mechanism, enabling context dependence and long-range dependencies between tokens. Recent work has drawn an equivalence between Hopfield networks, a kind of associative memory model, and Transformers. In this work, we leverage this bridge, using Vector Symbolic Architectures (VSA), a brain-inspired computational paradigm capable of representing and implementing data structures, including associative memory models, to define a broad class of attention mechanisms catered for complex data types. In particular, we use Generalized Holographic Reduced Representations (GHRR), an implementation of a VSA, as the foundation for our proposed class of attention mechanisms. We show that GHRR is capable of implementing attention and design a GHRR Transformer encoder architecture based on the demonstrated mathematical equivalence. We propose a new kind of binding-based positional encoding based on methods used in VSAs for encoding sequential information. We extend the attention mechanism in our architecture to support graphs, inspired by techniques used in VSAs to encode graph representations. We evaluate the GHRR Transformer on language modeling, vertex classification, and graph classification tasks. Results suggest that our approach provides benefits in language modeling and graph classification tasks compared to baseline models.

## 1 INTRODUCTION

The introduction of the Transformer Vaswani et al. (2017) brought about a revolution in AI, from language modeling to vision to reinforcement learning Brown et al. (2020); Dosovitskiy et al. (2021); Chen et al. (2021). Central to the Transformer architecture is the self-attention mechanism, enabling context dependence and long-range dependencies between tokens.

Recent work has drawn an equivalence between Hopfield networks model with a modified energy function Ramsauer et al. (2021); Hopfield (1982) and the self-attention mechanism. This equivalence is two-way and thus enables cross-pollination; unique features of one side can be transferred and applied to the other. For example, Ramsauer et al. (2021) applied the idea of repeated iteration of the Hopfield update rule to the self-attention mechanism within a transformer, which facilitates better memory retrieval. Consequently, one can use a similar strategy, exploiting the bridge between self-attention and associative memory models, e.g. Hopfield networks, to develop a broader class of self-attention mechanisms that can better handle data with more complex relations by utilizing associative memory structure.

In this work, we focus on Vector Symbolic Architectures (VSA), also known as Hyperdimensional Computing (HDC) Kanerva (2009); Kleyko et al. (2023), as a candidate associative memory framework for the extension of self-attention. While Hopfield networks rely on a dynamical update rule for memory retrieval, VSAs, as algebras of high dimensional vectors, are capable of performing associative memory operations purely based on their algebraic operators. That VSAs can perform the auto-associative capabilities of a Hopfield network and beyond motivates our choice for implementing and extending the attention mechanism.

In general, VSAs adhere to certain computational principles, including using high-dimensional vectors with holographic properties (i.e. the same information is present in each part of the representation, in expectation) and supporting the algebraic operations of bundling and binding, which correspond

to the cognitive operations of memorization and association, respectively. Any algebra satisfying these principles can be classified as a VSA. As a consequence, there exists a vast family of VSAs Gayler (1998); Plate (2003). In particular, in this work, we make use of Generalized Holograph Reduced Representations (GHRR) Yeung et al. (2024), an instantiation of a VSA to implement and extend self-attention. As will be apparent below, *the specific mathematical implementation of GHRR suggests a natural parallel to self-attention in the form of binding key, query, and value hypervectors*, i.e. the VSA equivalent of memory retrieval in a Hopfield network. More importantly, as an algebra, VSAs are endowed with natural compositional structure, which enables one to build complex data structures from simpler constituents without any change in representational space Kleyko et al. (2020); Frady et al. (2020); Kleyko et al. (2022); Poduval et al. (2022). *This property of VSAs, as we will develop in this work, provides a systematic method for one to extend self-attention to support more complicated data structures beyond sequences.* While VSAs traditionally follow stringent mathematical constraints, here, we make use of VSA as a conceptual framework for designing generalized self-attention mechanisms for structured data but allow for relaxation of the constraints. Our contributions are as follows:

1. We demonstrate the mathematical equivalence between the binding of key, query, and value hypervectors and the self-attention mechanism, i.e. that GHRR is capable of implementing self-attention.
2. Using the compositionality of VSAs and their ability to represent complex data structures, we extend the vanilla self-attention mechanism to one that naturally supports complex data types by construction.
3. We propose a new kind of binding-based positional encoding based on methods used in VSAs for encoding sequential information.
4. We verify our equivalence claim by evaluating a Transformer encoder with GHRR-based attention (hereon referred to as GHRR Transformer) on a language modeling task and compare it to a vanilla Transformer baseline.
5. We apply our methodology of extending self-attention using VSA principles to graph data and develop a GHRR Graph Transformer. We provide an interpretation of GHRR graph attention as performing a "soft" one-hop and evaluate our model on vertex and graph classification tasks.

## 2 RELATED WORK

**Transformer adaptation over structured data:** The transformer architecture has demonstrated remarkable versatility and has been adapted to handle a wide range of structured data types, including graphs Min et al. (2022), trees Wang et al. (2019), images Khan et al. (2022), time series Lim et al. (2021), and audio Verma & Berger (2021). In contrast to these works, which adapt the transformer to a specific data structure, our approach provides an adaptation framework for many data structures based on the structural representation of VSA; for this work, we adapt and experiment on the record-based encoding Imani et al. (2019); Ge & Parhi (2020) for text-based data and GrapHD Poduval et al. (2022) for graph-based data.

**Adaptation techniques for graph information:** When it comes to adaptation techniques, especially for graphs, there are three general methods for integrating graph information into the transformer Min et al. (2022): (1) injecting GNNs into transformer architectures as Auxiliary Modules, such as in Mesh Graphormer Lin et al. (2021) and Graph-BERT Zhang et al. (2020); (2) enhancing positional encoding (PE) with graph information, such as the Hop-based and Intimacy-based PE in Graph-BERT and centrality-based PE in Graphormer Ying et al. (2021), and (3) improving the attention matrix computation with graph information, techniques that include using graph kernels for attention Mialon et al. (2021) and adding soft bias to attention scores Zhao et al. (2021); Ying et al. (2021); Khoo et al. (2020). Our method for graphs takes the unique approach of altering the computation of the key matrix based on VSA graph representation, which subsequently alters positional encoding and attention computation.

**Transformers + VSA:** There exists some prior work leveraging VSA for Transformers. Peng et al. (2021) makes use of the Random Fourier Features (RFF) Rahimi & Recht (2007) encoding commonly used in VSAs Hernández-Cano et al. (2021) to efficiently compute attention weights in linear time and space. This work exploits the kernel approximation properties of a specific implementation of VSA. More recently, MIMOFormer Menet et al. (2024) leverages VSA's principle of computing in

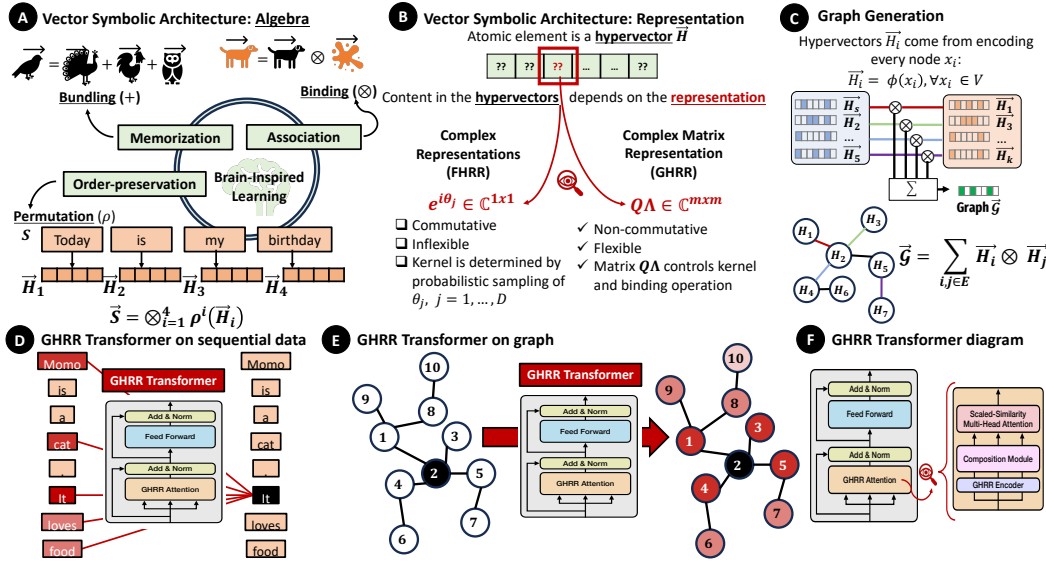

Figure 1: **A.** An overview of VSA operations, bundling, binding, and permutation, and their functional interpretations. **B.** A comparison of FHRR and GHRR VSA implementations. **C.** An example of how graphs can be encoded in VSAs. **D.** A visualization of how the GHRR Transformer performs attention on sequential data. **E.** A visualization of how the GHRR Transformer performs attention on graph data. **F.** An overview of the GHRR Transformer architecture.

superposition to provide a dynamic trade-off between model accuracy and throughput. In contrast to the above approaches which focus primarily on computation efficiency, our work focuses on building compositional representations with VSA's algebraic properties.

## 3 BACKGROUND

In this section, we first give a brief mathematical specification of the scaled dot-product attention mechanism typically used in a Transformer. We introduce VSAs, highlighting computational principles and basic algebraic operations and properties. Finally, we describe GHRR, a specific implementation of a VSA that we use in this work.

### 3.1 ATTENTION

We focus on scaled dot-product attention Vaswani et al. (2017), which can be written as $\text{attn}(Q, K, V) = \text{softmax}\left(\frac{1}{\sqrt{d_k}} Q K^\top\right) V$, where $Q, K, V$ are the embeddings of the input features corresponding to query, key, and value respectively, and $1/\sqrt{d_k}$ is the scaling factor determined by the embedding dimension $d_k$, which we omit for the rest of the paper. Generally, $Q, K, V$ is defined in terms of some input streams $X_q, X_k$, and $X_v$: $Q = X_q W_q$, $K = X_k W_k$, and $V = X_v W_v$. Then, we have

$$\text{attn}(Q, K, V) = \text{softmax}(X_q W_q W_k^\top X_k^\top) X_v W_v. \tag{1}$$

As suggested by the symbols $Q, K, V$, attention can be interpreted as querying a dictionary formed by associated keys and values.

### 3.2 VECTOR SYMBOLIC ARCHITECTURES

Vector Symbolic Architecture (VSA), also known as Hyperdimensional Computing (HDC), is a computing framework inspired by the brain. It is motivated by the observation that representations in the brain are high-dimensional, consisting of neural activations of a large population of neurons Kanerva (2009). Moreover, while these population-level representations appear to be highly distributed and

stochastic across different brains, they exhibit the same cognitive properties at a high level Kleyko et al. (2023); Gayler (1998).

The fundamental unit in a VSA is a high dimensional vector, also called a hypervector, corresponding to the population-level neural activations. A hypervector $H$ lives in some hyperspace $\mathcal{H}$, e.g., $\mathbb{R}^D$ for $D$ large. The collection of hypervectors, along with some operators, forms an algebra over vectors. Generally, there are two types of hypervectors: (1) base hypervectors, which are generated stochastically, e.g., $H \sim \mathcal{N}(0, I)$; and (2) composite hypervectors, which are created by combining hypervectors via the operators of the algebra. These hypervectors can be compared via a similarity function $\delta(H_1, H_2)$. Generally, base hypervectors are generated such that they are quasi-orthogonal with respect to the similarity function. The three main operations in VSA, bundling, binding, and permutation, can be characterized by how they affect the similarity of hypervectors. We describe the three operations below:

1. Bundling ($+$): Typically implemented as element-wise addition. If $H = H_1 + H_2$, then both $H_1$ and $H_2$ are similar to $H$. From a cognitive perspective, it can be interpreted as memorization.

2. Binding ($\odot$): Typically implemented as element-wise multiplication. If $H = H_1 \odot H_2$, then $H$ is dissimilar to both $H_1$ and $H_2$. Binding also has the important property of similarity preservation in the sense that for some hypervector $H_3$, $\delta(H_3 \odot H_1, H_3 \odot H_2) \simeq \delta(H_1, H_2)$. From a cognitive perspective, it can be interpreted as the association of concepts.

3. Permutation ($\rho$): Typically implemented as a rotation of vector elements. Generally, $\delta(\rho(H), H) \simeq 0$. Permutation is usually used to encode order in sequences.

It is important to note that the description above of VSA is general; there are various specific realizations of VSA with the above properties. Figure 1A illustrates the VSA operations and its interpretations.

Crucially, with the recursive application of the operations above, one can encode, represent, and query complex data structures such as sets, sequences, dictionaries, and graphs in the compressed form of a single hypervector Kleyko et al. (2023); Poduval et al. (2022).

### 3.3 Generalized Holographic Reduced Representations

GHRR is a specific implementation of a VSA. It is a generalization of the Fourier Holographic Reduced Representations (FHRR) framework Plate (2003). A GHRR base hypervector $H \in \mathbb{C}^{D \times m \times m}$ of dimension $D$ and complexity $m$ is of the form $H_j = W_j \Lambda_j$ for $j = 1, \ldots, D$. Here, $W_j$ is an $m \times m$ unitary matrix and $\Lambda_j$ is an $m \times m$ diagonal unitary matrix.[1] This is visualized in Figure 1B.

GHRR hypervectors are endowed with two operations, bundling and binding, which are defined by element-wise addition and matrix multiplication, respectively. We define the similarity between two hypervectors as $\delta(H_1, H_2) = \frac{1}{mD} \text{Re} \left[ \text{tr} \left( \sum_{j=1}^{D} H_{1j} H_{2j}^{\dagger} \right) \right]$, where $H_{1j}$ and $H_{2j}$ are the $j$-th matrix element of $H_1$ and $H_2$, respectively.

When not conditioned on an input, $\Lambda_j = \text{diag}(e^{i\theta_{j1}}, \ldots, e^{i\theta_{jm}})$, for $\theta_{jk} \sim p_k$ for distributions $p_k$ for $k = 1, \ldots, m$ such that $\mathbb{E}[e^{i\theta_k}] = 0$. It can be shown that this choice of $\Lambda_j$ and any arbitrary choice of unitary $W_j$ satisfies the constraints of a VSA given in section 3.2 Yeung et al. (2024).

Given some input $x \in \mathbb{R}^n$, we define $\Lambda_j(x) = \text{diag}(e^{iw_{j1}^{\top} x}, \ldots, e^{iw_{jm}^{\top} x})$, where $\mathbf{w}_{jk} \sim p_k$ with $p_k$ being symmetric distributions with zero mean. We denote a GHRR hypervector encoded in this way as $\phi(x) := [W_j \Lambda_j(x)]_{j=1}^D$. As in FHRR hypervectors using an RFF encoding scheme Rahimi & Recht (2007), $\delta(\phi(x), \phi(y))$ approximates a kernel, albeit a more complex one in the case of GHRR.

In general, one can interpret $\Lambda$ as the component primarily responsible for controlling the shape of the kernel, while $W$ controls how hypervectors bind together. This is in contrast to prior VSA implementations, which lack expressivity in the binding operation, further accentuated by the fact that GHRR uses matrix multiplication, as opposed to scalar multiplication, for the binding operation.

---

[1] Although it is more proper to describe $H_j$ as a "hypermatrix" or "hypertensor", we stick to the term hypervector both by convention and by the understanding that each component of a GHRR base hypervector is a single element of the unitary group of degree $m$, which happens to be representable by matrices.

Table 1: Table of mathematical symbols

| Symbol | Description |
|---|---|
| $D \in \mathbb{Z}_{\geq 1}$ | Hyperdimension; number of heads |
| $m \in \mathbb{Z}_{\geq 1}$ | Complexity; embedding dimension per head |
| $n \in \mathbb{Z}_{\geq 1}$ | Input dimension |
| $Q \in \mathbb{C}^{D \times m \times m}$ | Query hypervector |
| $K \in \mathbb{C}^{D \times m \times m}$ | Key hypervector |
| $V \in \mathbb{C}^{D \times m \times m}$ | Value hypervector |
| $W \in \mathbb{C}^{D \times m \times m}$ | Weight component of GHRR hypervector |
| $\Lambda \in \mathbb{C}^{D \times m \times m}$ | Exponential random diagonal matrix component of GHRR hypervector |
| $\phi : \mathbb{R}^n \to \mathbb{C}^{D \times m \times m}$ | GHRR encoder |
| $P \in \mathbb{C}^{D \times m \times m}$ | Positional encoding hypervector for binding-based positional encoding |

Taken together, the use of matrix multiplication for binding and the ability to modulate $W$ allows GHRR to be integrated more naturally into a connectionist model like the Transformer.

## 4 GHRR AND ATTENTION

### 4.1 GHRR IMPLEMENTS ATTENTION

We first show that it is possible to match the mathematical form of the attention mechanism given in Eq. 1 using GHRR. We then show that, by binding specific positional information to GHRR token hypervectors, we can implement token-level attention using GHRR.

**Matching mathematical forms between GHRR and attention.**   Suppose we have three hypervectors $Q = W_q \Lambda_q$, $K = W_k \Lambda_k$, and $V = W_v \Lambda_v$. Then we can write

$$[\text{softmax}(\text{Re}[QK^\dagger])V]_j = \text{softmax}(\text{Re}[W_{qj}\Lambda_{qj}\Lambda_{kj}^\dagger W_{kj}^\dagger])W_{vj}\Lambda_{vj}. \qquad (2)$$

The almost-equivalence between Eq. 1 and Eq. 2 suggests that GHRR is capable of implementing attention. The GHRR representation enforces unitarity on $W_j$ to generate base hypervectors that are norm-preserving; we relax this constraint for greater flexibility, enabling expressivity comparable to that of a traditional Transformer.

While traditional attention is applied to a sequence of tokens as encoded by a matrix $X$, here, this is not necessarily the case. The analog of $X$, $\Lambda$, is a diagonal matrix that in general encodes only one token. Thus, the attention described in Eq. 2, while similar in form to Eq. 1, does not implement attention over tokens; instead, it applies attention over the representation of a single token. We distinguish this form of *representation-level attention* from traditional *token-level attention* which is explicitly applied only to token representations.

**Token-level attention and beyond via VSA positional encoding.**   To add token-level information, one can express the hypervectors as a sum of token hypervectors bound with hypervectors encoding positional information. To simplify notation, let us consider only one dimension of the hypervector, allowing us to omit and free up the subscript $j$ on $W$ and $\Lambda$ for GHRR components. More precisely, let $\phi(x) = W\Lambda(x)$ be the GHRR encoding and $E_j$ be the positional information for the $j$-th token. In particular, we let $E_j$ be an $m \times m$ matrix such that $[E_j]_{jj} = 1$ and zero everywhere else. Let $x_1, \ldots, x_m$ be the tokens we wish to encode. Then our sequence encoding is

$$\phi(x_1, \ldots, x_m) := \sum_{j=1}^{m} E_j \phi(x_j), \qquad (3)$$

which results in a matrix where exactly the $j$-th row encodes information about the $j$-th token. This construction gives an explicit correspondence between GHRR and attention, with the difference being that GHRR token representations involve an extra random exponential map as determined by $\Lambda$.

In addition, we may let $E_j$ be some arbitrary trainable matrix $P_j$, which essentially makes it a learnable position encoding present in each transformer block. In contrast to the sinusoidal positional encoding commonly used in the vanilla Transformer architecture, which is added to the

token embeddings prior to the application of any Transformer blocks, this *binding-based positional encoding* is applied via the binding operation, i.e. matrix multiplication in GHRR, and is included in every GHRR Transformer block.

**Matching parameters between GHRR and Transformer.** Given the above discussion, we propose to treat each component of a GHRR hypervector as an attention head. In other words, the hyperdimension $D$ in GHRR corresponds to the number of attention heads, and the complexity $m$ determines the maximum number of tokens the GHRR implementation of attention can deal with independently. Note that it's possible to encode more than $m$ tokens at the cost of them entangling within the representation. For simplicity in analysis, we do not consider the entangled case for this work; as a result, the range of attention is limited by $m$.

## 4.2 STRUCTURING TRANSFORMERS EMBEDDING WITH GHRR OPERATIONS

Given GHRR's ability to perform attention, we can naturally replace the attention mechanism with the GHRR equivalent as described in Section 4.1. Moreover, our formulation suggests a way in which the attention mechanism can be extended.

Due to GHRR's holographic nature, one can encode more complex representations within a tensor of the same shape. This suggests an extension of the GHRR attention module where query, key, and value hypervectors themselves can be composites depending on the nature of the data. In the simple case of sequential input as in a vanilla transformer, and as described in Section 4.1, a sequence can be represented in the form $\sum_j \mathbf{p}_j \odot \mathbf{x}_j$, which resembles Eq. 3.

If, instead, the data is from a more complex data structure, e.g. a graph, we can use a corresponding VSA encoding that reflects its structure, as shown in Figure 1C. *In particular, the key and query can be formulated in a way that reflects the way is done in traditional VSA applications, e.g. querying for the neighbors of a vertex in a graph Poduval et al. (2022).* This approach to encoding structure is *global* in the sense that the encoding is constructed to explicitly capture the entire data structure in question; it is built into the model architecture based on prior knowledge of the data. Figure 1F illustrates the general encoder block structure as well as the structure of the attention module. The general encoding block structure is analogous to that in a vanilla Transformer, with GHRR attention instead of the traditional scaled dot-product attention.

## 5 TECHNICAL DETAILS

### 5.1 GHRR TRANSFORMER FOR SEQUENCES

Let $x_1, \dots, x_n$ be a sequence. Without loss of generality, we assume $D = 1$, so all hypervectors are simply $m \times m$ matrices. Moreover, we assume $m = n$. Let $\phi_q, \phi_k, \phi_v$ be query, key, and value encoders respectively. We define the hypervectors $Q, K, V$ as follows:

$$Q = \sum_{j=1}^{n} P_j^q \odot \phi_q(x_j), \quad K = \sum_{j=1}^{n} P_j^k \odot \phi_k(x_j), \quad V = \sum_{j=1}^{n} P_j^v \odot \phi_v(x_j), \tag{4}$$

where $P_j^q, P_j^k, P_j^v$ for $j = 1, \dots, n$ are positional encoding matrices. With $D = 1$, binding reduces to matrix multiplication. Figure 2A illustrates the general architectural diagram.

### 5.2 GHRR TRANSFORMER FOR GRAPHS

Let $G = (\mathcal{V}, \mathcal{E})$ be an undirected graph where the vertices are labeled but the edges are not. We assume that the parameter $m = |\mathcal{V}|$ for our GHRR encoding and let $o : \mathcal{V} \to \{1, \dots, m\}$ be a bijection mapping each vertex to an index. Without loss of generality, we assume $D = 1$, so all hypervectors are simply $m \times m$ matrices. We encode query, key, and value hypervectors, $Q, K, V$, respectively, as follows:

$$Q = \sum_{x \in \mathcal{V}} P_{o(x)}^q \odot \phi_q(x), \quad V = \sum_{x \in \mathcal{V}} P_{o(x)}^v \odot \phi_v(x), \tag{5}$$

$$K = \sum_{(u,v) \in \mathcal{E}} (\phi_{k2}(v) \odot P_{o(v)}^k)^\dagger \odot \phi_{k1}(u), \tag{6}$$

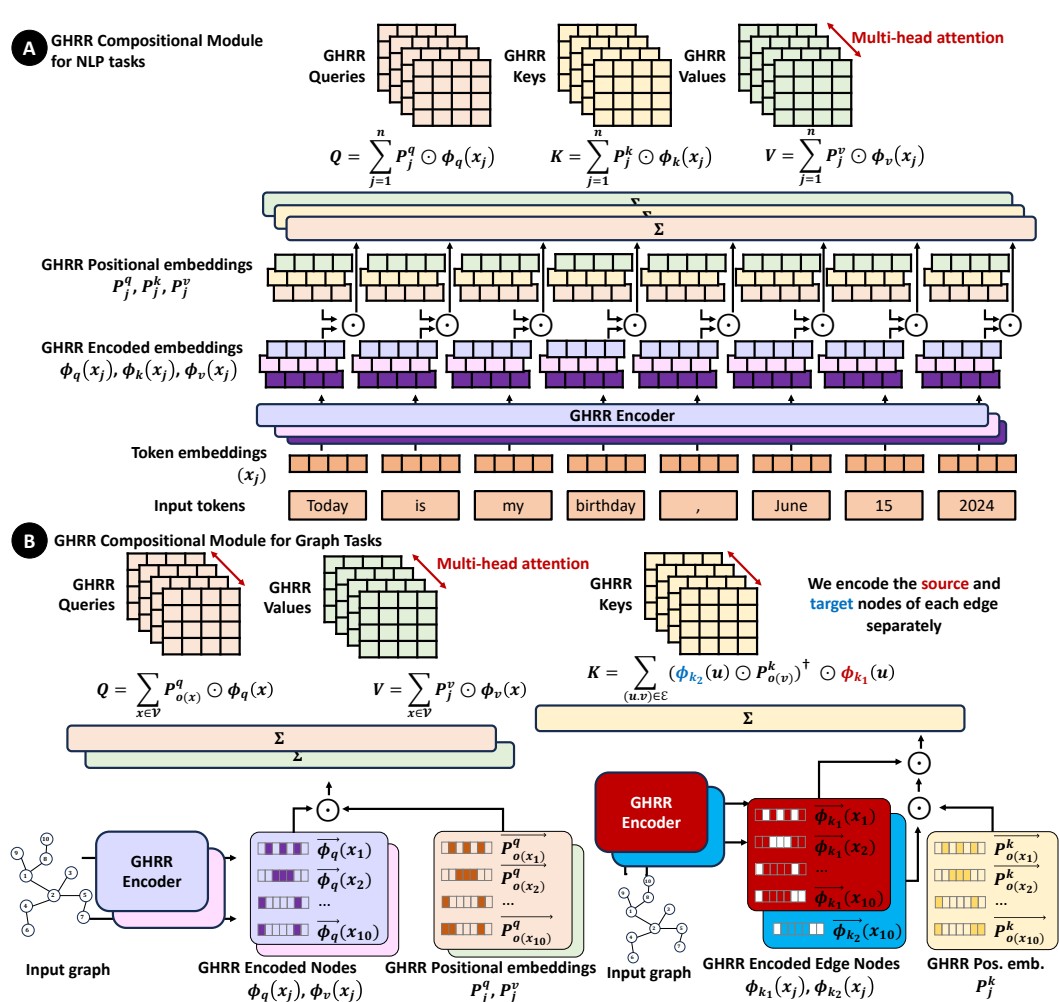

Figure 2: **A.** A visualization of the architecture of the compositional module for the sequential GHRR Transformer. **B.** A visualization of the architecture of the compositional module for the GHRR Graph Transformer. Compared to the sequential version, the key hypervector is computed differently.

where $P_j$ for $j = 1, \ldots, m$ are positional encoding matrices. This method of representing a graph is similar to that in GrapHD Poduval et al. (2022). When computing the attention weights (before softmax), we get

$$QK^\dagger = \sum_{x \in \mathcal{V}} \sum_{(u,v) \in \mathcal{E}} P^q_{o(x)} \phi_q(x) \phi_{k1}(u)^\dagger \phi_{k2}(v) P^k_{o(v)}. \tag{7}$$

We show that with some assumptions and a careful choice of encoding matrices in $\phi_q, \phi_{k1}, \phi_{k2}, \phi_v, P^q_j, P^k_j$, the graph GHRR self-attention can be interpreted as a soft one-hop.

We first assume that $P^q_j = P^k_j = P^v_j = E_j$, where $E_j$ is defined above in subsection 4.1. In addition, we make the simplifying assumption that $\phi_{k2}(v) \in \mathbb{C}^{m \times m}$ is a fixed matrix of all ones and denote the $l$-th row of $\phi_q(x), \phi_{k1}(u)$ as $q_l(x), k_l(u)$ respectively, for $l = 1, \ldots, m$. Then,

$$[E_{o(x)} \phi_q(x) \phi_{k1}(u)^\dagger \phi_{k2}(v) E_{o(v)}]_{ij} = \delta_{io(x)} \delta_{jo(v)} q_i(x) \sum_{l=1}^{m} k_l(u)^*, \tag{8}$$

where $\delta_{ij}$ is the Kronecker delta symbol. If we interpret $e(x) := q_j(x)$ and $e'(u) := \sum_{l=1}^{m} k_l(u)$ as vertex embeddings, $s(x, y) := e(x)e'(y)^*$ defines an asymmetric similarity measure between vertices $x, y \in \mathcal{V}$, which can be made symmetric by weight sharing and an additional assumption. Then, by Eq. 7,

$$[QK^\dagger]_{ij} = \sum_{x \in \mathcal{V}} \delta_{io(x)} \sum_{(u,v) \in \mathcal{E}} \delta_{jo(v)} s(x, u) = \sum_{u \in N_G(o^{-1}(j))} s(o^{-1}(i), u), \tag{9}$$

where $N_G(x)$ is the set of neighbors of $x \in \mathcal{V}$.

For the sake of illustration, let us suppose that $s$ is the discrete metric on $\mathcal{V}$. Then $[QK^\dagger]_{ij} = \mathbb{I}[o^{-1}(i) \in N_G(o^{-1}(j))]$, where $\mathbb{I}$ is the indicator function. If we denote the $j$-th row of $V$ as $v_j$, which encodes vertex $o^{-1}(j)$ due to the positional encoding matrix, we have the result

$$\left[\text{softmax}(\text{Re}(QK^\dagger))V\right]_i = \frac{1}{|N_G(o^{-1}(i))|} \sum_{x \in N_G(o^{-1}(i))} v_{o(x)}. \tag{10}$$

Thus, after applying GHRR graph attention, the new representation of vertex $i$ consists of the sum of the previous representations of its neighbors, which is exactly a one-hop. Figure 1E provides intuition for how attention is applied after multiple applications of the Graph Transformer Encoder block. Figure 2B illustrates the general architectural diagram.

# 6 RESULTS

## 6.1 NEXT TOKEN PREDICTION

We evaluate our model on a next-token prediction language modeling task on the Wikitext2 Merity et al. (2016) and the Penn Treebank Marcus et al. (1993) datasets. We implement a Transformer encoder with GHRR attention as described in section 5.1.

The positional encoding can be the same or different across the $Q, K, V$ hypervectors as well as across attention heads. Moreover, the positional encodings can either be trainable or fixed (i.e. randomly initialized). This gives us eight different variants of the GHRR Transformer model. For each GHRR encoder $\phi_q, \phi_k, \phi_v$, we make $W$ trainable and keep $\Lambda$ fixed. Sample positional encodings are visualized in Appendix C.

Both the baseline Transformer model Vaswani et al. (2017) and our model have a comparable number of weight parameters, with a slight increase when trainable positional encoding (PE) is included. Training details are described in Appendix B.1. We report the mean perplexity (PPL) and standard deviation over five independent runs in Table 2.

We observe an average performance improvement of 5.47% on WikiText-2 and 2.75% on the PTB dataset when compared to the baseline Transformer model. In the cases with the highest observed

Table 2: Perplexity of trained language models

| Model | Trainable PE | Wikitext2 Merity et al. (2016) | Penn Treebank Marcus et al. (1993) |
|---|---|---|---|
| Baseline | N/A | $29.16 \pm 0.13$ | $94.78 \pm 0.23$ |
| All same | No | $\mathbf{27.54 \pm 0.08}$ | $93.93 \pm 0.09$ |
| QKV | No | $27.6 \pm 0.10$ | $91.63 \pm 0.04$ |
| Head | No | $27.55 \pm 0.06$ | $91.68 \pm 0.11$ |
| All different | No | $27.56 \pm 0.09$ | $91.58 \pm 0.08$ |
| All same | Yes | $\mathbf{27.54 \pm 0.08}$ | $93.98 \pm 0.12$ |
| QKV | Yes | $27.57 \pm 0.12$ | $\mathbf{91.51 \pm 0.08}$ |
| Head | Yes | $27.56 \pm 0.07$ | $91.53 \pm 0.09$ |
| All different | Yes | $27.56 \pm 0.05$ | $91.52 \pm 0.05$ |

Table 3: Vertex classification accuracy

| Model | Accuracy (%) |
|---|---|
| GHRR Graph Transformer | 82.30 |
| GRIT Ma et al. (2023) | 87.20 |
| EGT Hussain et al. (2022) | 86.82 |

Table 4: Graph classification accuracy

| Trainable PE | Accuracy (%) |
|---|---|
| No | 53.5 |
| Yes | 70.0 |

improvements, the performance increased by 5.53% on WikiText-2 and 3.44% on PTB, respectively. Specifically, we found a 2.56% performance improvement in PTB when PEs are varied across $Q, K, V$ matrices, attention heads, or both, compared to when they were not.

This suggests that the inclusion of PEs help with model performance, though there needs to be some kind of variation between PEs to have sufficient expressive power. Moreover, there is negligible difference between fixed and trainable positional encodings, suggesting that level of expressive power is not needed for this task.

## 6.2 VERTEX CLASSIFICATION

We evaluate GHRR attention on vertex-level tasks of graph pattern recognition using the PATTERN dataset. To reduce model parameters, we apply the assumptions from Section 5.2, including fixed positional encodings and rank-1 GHRR hypervectors, resulting in a model that performs one-hop attention. Detailed training information and dataset description are provided in Appendix B.2.

The accuracies of the GHRR graph transformer and state-of-the-art (SOTA) algorithms are listed in Table 3. Although the performance of the GHRR graph transformer is slightly lower than that of the SOTA, it demonstrates a significant advantage in training efficiency. The GHRR graph transformer converges after just 3 epochs, whereas the other models require tens of epochs to converge.

## 6.3 GRAPH CLASSIFICATION

To test the efficacy of our model, we use a synthetic graph classification dataset. The model we use is a GHRR Graph Transformer with positional encodings that are distinct across $Q, K, V$ hypervectors but are the same across attention heads. We do not make the assumptions in Section 5.2. Table 4 compares the accuracy of two versions of the model: one where positional encodings are fixed and one where they are trainable. Results suggest that unlike in the language modeling task, trainable positional encodings provide a significant advantage in model performance. Training details are placed in Appendix B.3.

## 7 DISCUSSION

**Attention based on other VSA structure encodings.** In this work, we explored attention mechanisms based on VSA encodings of sequences and graphs. Of course, our approach is not just

limited to these two data structures; one can design a corresponding GHRR Transformer for every kind of VSA data structure encoding, including that for trees Frady et al. (2020) and finite state machines Kleyko et al. (2023).

**Positional encoding extensions.** Our proposed positional encoding based on the binding operation in VSA depends solely on the index of the token. More complex variants of the encoding can be explored; for example, one can consider a 2D positional encoding based on $(x, y)$-coordinates in images $P_x \odot P_y$. As mentioned in the related work, one can also incorporate graph information Ying et al. (2021) into the positional encodings.

**Decoder Architectures.** While we only considered a Transformer encoder architecture in this work, our formulation extends naturally to a decoder model. One consideration when designing the decoder model is how sequential generation can occur, given data of more complex types. Efficient attention masking is also of practical concern when designing the decoder architecture.

## 8 CONCLUSION

We have shown that among VSAs, GHRR is capable of implementing the attention mechanism in the form of the binding operation, and develop a GHRR Transformer based on this equivalence. We introduced a novel binding-based positional encoding and extended the attention mechanism to support complex data structures based on VSA principles. In particular, as an example, we specify a graph transformer architecture based on our framework and provide an interpretation for the graph GHRR attention mechanism as performing a one-hop. We then evaluate our GHRR Transformer variants on language modeling, vertex classification, and graph classification tasks.

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

## A  HOPFIELD NETWORKS

A Hopfield network is an auto-associative memory model; i.e. it retrieves memory items based on the content of the memory itself (i.e. the keys and values are the same). It can be implemented as a neural network that stores patterns $\{\xi_j\}_{j=1}^n$ such that $\xi_j \in \{-1, 1\}^d$ are attractors Hopfield (1982). An input query $\xi$ is passed into the network. The retrieved vector is determined by the update rule

$$\xi_{t+1} = \text{sgn}(XX^\top \xi_t) \tag{11}$$

where $X = [\xi_1, \ldots, \xi_n]$. Under conditions such as sufficient separability of patterns (with respect to dot product) and $n$ being sufficiently small, $\xi$ will converge to the closest pattern $\xi_j$. The capacity of the Hopfield network is $O(d)$.

Ramsauer et al. (2021) extends the network to continuous states and introduces a new update rule that is equivalent to the self-attention mechanism:[2]

$$\xi_{t+1} = X\text{softmax}(\beta X^\top \xi_t) \tag{12}$$

Here, the patterns $\{\xi_j\}_{j=1}^n$ satisfy $\xi_j \in \mathbb{R}^d$. Compared to the discrete network defined, the resulting Hopfield network has exponential storage capacity.

Hopfield networks, as in attention, compute pairwise similarities across the entire set of inputs, performing simple dictionary-like queries. For more complex inputs, however, we might wish to encode structural information into the representation so as to achieve a more structure-aware query operation.

## B  TRAINING DETAILS

All experiments are conducted on a workstation equipped with an AMD Ryzen Threadripper PRO 5965WX CPU and two NVIDIA GeForce RTX 4090 GPUs. Each GHRR layer consumes approximately 0.5 GB of VRAM. The model requires approximately 70 minutes to execute one epoch for vertex classification and 10 minutes for one epoch for graph classification.

### B.1  NEXT TOKEN PREDICTION

For the use case of next token prediction, we add a few constraints to enable easy causal masking for computational efficiency. We use causal masking to prevent the model from "looking ahead" when making next token predictions. In particular, we set the positional encoding matrices $P_j^q, P_j^k, P_j^v$ to have the form $P_j^a = E_j A_j^a, a \in \{q, k, v\}$. Doing so confines information about the $j$-th token to the $j$-th row in the $Q, K, V$ matrix elements, which allows us to apply a causal mask on the matrix $\text{Re}(QK^\dagger)$.

The models are trained using the Adam optimizer with a learning rate of 1e-3 and a weight decay of 1e-3 over 20 epochs. The baseline model features an embedding size of 240, a hidden dimension of 200, and a dropout rate of 0.2, while our model uses an embedding dimension of 240, distributed across 8 heads.

### B.2  VERTEX CLASSIFICATION

The PATTERN dataset is widely utilized to model social network communities by modulating intra- and extra-community interactions Dwivedi et al. (2023). It comprises 10,000 training graphs, 2,000 validation graphs, and 2,000 test graphs. The graphs within this dataset are generated using the Stochastic Block Model (SBM) Abbe (2018). An SBM is a type of random graph where communities are assigned to each node. In this model, any two vertices are connected with a probability $p$ if they belong to the same community, or with a probability $q$ if they belong to different communities, where $q$ represents the noise level.

The models comprise 8 GHRR attention layers and are trained using the Adam optimizer with an initial learning rate of 1e-6, which decreases by a factor of 0.2 after 5 epochs without improvement. Each model has an embedding size of 1880, a dropout rate of 0.2, and 10 attention heads.

---

[2]Self-attention has additional linear maps applied to the matrix $X$ to compute $Q, K, V$.

## B.3    GRAPH CLASSIFICATION

The synthetic dataset consists of random undirected graphs, each with 32 vertices and a corresponding binary label denoting whether the graph is fully connected or not. Graphs are sampled by randomly generating matrices $A'$ such that $P(A'_{ij} = 1) = p = 0.06$ for all $i, j = 1, ..., n$, where $n = 32$ is the number of vertices. We compute the final adjacency matrix as $A = \min(A' + (A')^\top, \mathbf{1})$, where $\mathbf{1}$ is a matrix of all ones. We synthesize 10,000 graphs for the training set, and 2,000 graphs each for the validation and test sets, respectively.

The models comprise 4 GHRR attention layers and are trained using the Adam optimizer with an initial learning rate of 1e-3, reduced by a factor of 0.5 after 2 epochs of plateau. Each model has an embedding size of 320, a dropout rate of 0.2, and 10 attention heads.

# C    POSITIONAL ENCODINGS

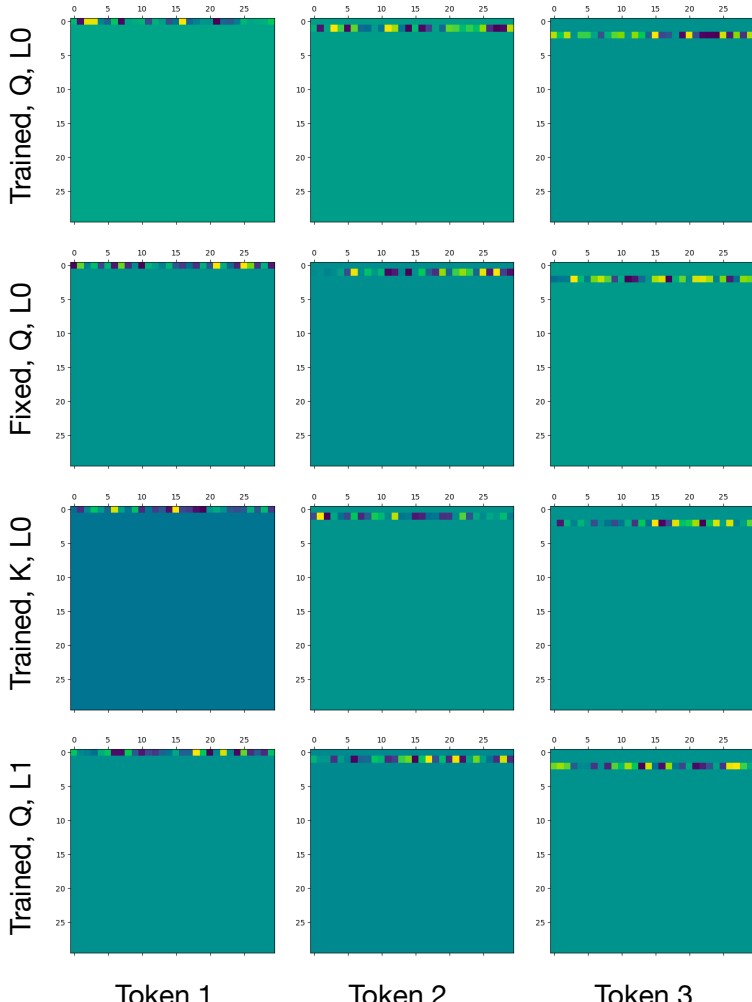

Figure 3: Visualization of positional encodings for the language modeling task, including trained positional encodings for $Q, K$ on two different Transformer layers and a fixed positional encoding for $Q$ in layer 0.

