# OpenReview forum: "Structure-aware Attention based on Vector Symbolic Architectures"
_ICLR.cc/2025/Conference — Submitted to ICLR 2025_

### Official Review · Reviewer_mr6j · 2024-11-01

**Soundness:** 2
**Presentation:** 1
**Contribution:** 1
**Rating:** 3
**Confidence:** 4

**Summary:**

The work discusses how a specific implementation of Vector symbolic architectures (VSA) can be interpreted as a multi-head attention model for compound data encoding. The manuscript shows under which restrictions of the VSA model its computation map to self-attention with fixed context window. It further provides a discussion on how to define specialized positional encodings to process sequential and graph structured data. Experiments are provided on simple next token prediction, node classification and graph classification datasets.

**Strengths:**

*  The idea of deriving a more general framework for self-attention through VSA is interesting. In particular as it may enable a principled way to manipulate the inductive bias of the attentional module by operating on how the underlying vector space is generated and how positional information is embedded in the representation by the algebra operators. This makes the work, in principle, able to achieve some impact on the community.

 *  The proposed approach, surprisingly, does not imply a substantially higher parameterization and computational costs (at least according to what the Authors state in the paper, as there is no specific experimental data supporting this claim).

**Weaknesses:**

* The work appears rather derivative when coming down to the learning methodology. The model underlying the proposed GHRR Transformer derives largely from Yeung et al 2024, aside from the part which integrate positional encoding information in the neural representation. Overall, the proposed equivalence of GHRR to self-attention boils down to a resemblance of the matrix operations involved in self-attention, when substantial constraints are imposed to the GHRR model. There is really no deep study and assessment of the relationship between the GHRR Transformer and self-attention, and of what are the consequences of some design choices and simplifications introduced in GHRR. Without a deeper insight on the key advantages introduced by the proposed GHRR-Transformer/Self-attention equivalence, this may appear rather empty and mostly a technical exercise.  The equivalence itself hinges on quite strong simplifying assumptions, which are mentioned but whose impact is not discussed in depth.  For instance, GHRR assumes a fixed context window: if such assumption is relaxed, then the dimensions of the embedding becomes entangled, loosing the very motivation for having introduced an holographic representation in a first instance. This aspect is mentioned in the paper, but only marginally, while it seems a major limitation of the approach.

* The work appears derivative also with respect to the contribution on graph processing. The encoding of graph data in the proposed GHRR Transformer is heavily based on Poduval et al, 2022. The paper is not clear about what novel insight is being provided on graph encoding by the proposed approach against the one of GraphHD. It would be helpful if the Authors could elaborate more on what are the novelties of the proposed approach agains GraphHD. The technical discussion on the properties of the GHRR Transformer against what is the state of the art in graph NNs, remain on  a shallow level. For instance, if I got this right, the disentanglement properties of the proposed Holographic embedding should allow to memorize exact information about inter-nodes relationships in the vertex encoding. This may be relevant with respect to the literature discussion about how to design graph NNs capable of capturing long range node relationships, surpassing limitations such as oversquashing. The work appears not well positioned with respect to relevant literature in this respect.

 * The work, at some point, relaxes the assumption on W being unitary. My understanding is that such an assumption in needed to preserve the holographic nature of the embeddings. It cannot be relaxed without a proper discussion of how this affects the properties of the model (the discussion should be both theoretical and empirical in this sense). Taking assumptions which contradict the very fundamental reasons for having introduced the holographic approach in the first instance, reduces the soundness of the contribution.

 * The empirical analysis is very limited in scope, depth and reproducibility. Little details are provided as concerns the experimental setup  and no reference to code is given (neither public anonimyzed nor attached to the submission as supplementary). It would be helpful if the Authors can provide additional details to facilitate reproducibility, including choice of optimizers, hyperparameters, as well as to gain a deeper insight into the computational charateristics of the approach, such as its computational costs and parameterization in the experiments.

* The experiments on sequential data are too limited: only a single 2-dataset experiment with simple next token prediction tasks is provided. If the approach is put forward as an holographic equivalent of Transformers (in year 2024), then one would expect to see experiments on how the approach can be used to at least match a Transformer on proper language modeling tasks. The experiments with graph data are quite poor. There is no reference baseline model from literature (I would have expected a comparison with at least the most popular Graph Transformer models). The datasets used in the experiments are not widely recognized benchmarks by the graph NN community and this does not allow to compare the proposed model against the relevant related literature. The dataset on graph classification cannot be considered a proper graph benchmark: deciding between a fully connected and a non-fully connected graph does not require a model with the ability to capture complex structured relationships in a graph. It suffices a model which can count the number of ones in the adjacency matrix.

**Questions:**

1) Can you please provide evidence of what happens empirically when the model is used with a token context window n larger than m?

2) Can you elaborate on the effect of relaxing unitarity of W?

3) Can you provide experiments on widely accepted graph benchmarks, such as those on the Open Graph Benchmark (OGB)?

---

### Official Review · Reviewer_hif3 · 2024-11-01

**Soundness:** 3
**Presentation:** 3
**Contribution:** 3
**Rating:** 6
**Confidence:** 2

**Summary:**

The authors present a theoretical bridge between Vector Symbolic Architectures and transformer attention through the specific VSA of Generalized Holographic Reduced Representations. They show how the transformer attention mechanism can be derived as a specific formulation of the GHRR with some slight relaxations. They then show that their derivation is correct by implementing it and comparing against vanilla transformers in a few settings. They also show how the VSA formulation more naturally extends to graph learning.

**Strengths:**

- referred to bridge between transformers and hopfield networks
- presented material in a digestible way even for those who have never heard of VSAs
- demonstrate a formal, theoretical unification between VSAs and transformer attention
- support theoretical claims with empirical evidence
- show that their formulation can be readily used in another domain (graph classification)

**Weaknesses:**

- as I have never encountered VSAs before, I'm unsure about impact/significance of VSAs in general. Does the broader ICLR audience care about VSAs? Why should they care?
- difficult to understand the variants presented in Table 2. It would be nice if the differences between the variants was more clearly summarized closer to the point at which they are presented. Were they described in the methods?
- would have appreciated a comparison of the graph and vertex classification results to vanilla transformers
- generally difficult to understand some of the notation (probably due to my lack of exposure to VSAs)

**Questions:**

- unclear to me why there was a significant performance improvement over transformers if the attention mechanism is theoretically the same. Could you perhaps elaborate on the specific differences in the GHRR transformers that could contribute to the performance differences in the discussion/results? Ideally provide some analyses to causally investigate the important differences.
- why does the VSA formulation help with graphs in a way that vanilla transformers cannot already do? Graphs can readily be encoded as tokens in a context.

---

### Official Review · Reviewer_pY3Z · 2024-11-03

**Soundness:** 3
**Presentation:** 2
**Contribution:** 3
**Rating:** 3
**Confidence:** 3

**Summary:**

The paper proposes a new transformer architecture based on vector symbolic architectures (VSAs). The paper uses the Generalised Holographic Reduced Representation implementation of VSAs, which represents tokens as complex hypervectors ($\mathbb{C}^{D\times m \times m}$), and models interactions between these through bundling and binding algebraic operators.
The paper implements a transformer for sequences, and a transformer for graphs using GHRR, and demonstrates experimental results on language modelling and vertex classification tasks.

**Strengths:**

- The paper considers an interesting goal: implementing transformers using GHRR, which has rich algebraic structure
- A clear and informative description of VSAs and GHRR is provided.
- The proposed model is original; both for sequences and graphs.
- Valuable experimental results are demonstrated

**Weaknesses:**

- The presented results for the language modelling task use a single baseline, the vanilla transformer. For this model, the chosen embedding size is quite small, compared to those in the literature. It is also uncommon that the hidden dimension for the transformer is smaller than the model dimension.
- No hyperparameter search is described for either task.
- The presented results thus are not entirely informative about the model's performance.
- While VSA's are known to have rich algebraic structure, the paper does not discuss this for the presented model. It would have been valuable to demonstrate whether, for example, semantically similar words are mapped to GHRR representations with high similarity as measured by $\delta$ described on line 170. This would serve to motivate the architecture better.
- Related to the above, it is not clear from line 242 whether the GHRR version of attention uses the similarity measure from the VSA, which would be most natural.

**Questions:**

- Is the $\delta$ similarity measure used in the model itself?
- How were the hyperparameters for model training chosen?

---

### Official Review · Reviewer_76E9 · 2024-11-04

**Soundness:** 2
**Presentation:** 3
**Contribution:** 3
**Rating:** 3
**Confidence:** 3

**Summary:**

This paper proposes a Transformer encoder architecture based on Generalized Holographic Reduced Representations (GHRR), a Vector Symbolic Architecture (VSA) paradigm capable of implementing data structures including attention mechanisms. The proposed architecture uses binding-based positional encoding to encode sequential data, and also supports graph inputs. The architecture is evaluated on language modeling as well as node and graph classification tasks, showing better perplexity on language modeling tasks and comparable performance on vertex classification.

**Strengths:**

1. The authors draw useful connections between self-attention based Transformer models and Vector Symbolic Architectures, and use it to derive mathematical equivalence between QKV hypervectors and the self-attention mechanism.
2. The authors use the insights to construct self-attention mechanisms for more complex data, proposing a GHRR-based Graph Transformer architecture for graph inputs.

**Weaknesses:**

- The benchmarking of GHRR is limited; next token prediction perplexity is reported on two language datasets in Table 2 against a vanilla transformer, however the language modeling results would be strengthened by benchmarking on more recent NLP benchmarks and tasks, for instance the LAMBADA dataset for natural language understanding. Performance on a node classification and graph classification task is reported in Tables 3 and 4, however both experiments are missing baseline graph transformer models such as GPS Graph Transformer [1] and Graphormer [2]. Evaluating on more standard GNN benchmark datasets, such as ZINC and Open Graph Benchmark datasets, would also strengthen the empirical results of GHRR.

1. Rampášek, Ladislav, et al. "Recipe for a general, powerful, scalable graph transformer." Advances in Neural Information Processing Systems 35 (2022): 14501-14515.
2. Ying, Chengxuan, et al. "Do transformers really perform badly for graph representation?." Advances in neural information processing systems 34 (2021): 28877-28888.

**Questions:**

How does the runtime of GHRR compare to that of the vanilla Transformer? Given that vanilla attention has quadratic complexity, it would be good to know the complexity of GHRR relative to the standard self attention mechanism.

---

### Meta-Review · Area_Chair_cKus · 2024-12-16

**Metareview:**

This paper proposes a new transformer architecture but the evaluation is lacking.

While the reviewers generally found the connection of Transformers with vector symbolic architectures well-presented, they found the experimental evaluation lacking. For example, benchmarking on more datasets (76E9, mr6j), better choice of hyperparameters for the baseline (pY3Z, mr6j) would improve the paper. The motivation behind VSA’s is also unclear (pY3Z, hif3).

**Additional Comments On Reviewer Discussion:**

-

---

### Decision · Program_Chairs · 2025-01-22

Reject